

# Reliability and validity of an equanimity questionnaire: the two-factor equanimity scale (EQUA-S)

Catherine Juneau[1], Nicolas Pellerin[2], Elliott Trives[3], Matthieu Ricard[4], Rébecca Shankland[5] and Michael Dambrun[1]

[1] LAPSCO CNRS UMR 6024, Université Clermont Auvergne, Clermont-Ferrand, France
[2] CLLE-LTC CNRS UMR 5263, Université de Toulouse-le-Mirail (Toulouse II), Toulouse, France
[3] LAPCOS, Université de Nice-Sophia Antipolis, Nice, France
[4] Mind and Life Institute, Shechen Monastery, Kathmandu, Nepal
[5] LIP/PC2S, Université Grenoble Alpes, Grenoble, France

Corresponding authors
Catherine Juneau,
catherine.juneau@uca.fr
Michael Dambrun,
michael.dambrun@uca.fr

## ABSTRACT

**Background:** Many studies have revealed the positive impact of mindfulness training on mental health and proposed equanimity as a general outcome in contemplative research. Despite recent interest, relatively few studies have empirically examined equanimity and measurement instruments are still lacking. The main goal of this study was to develop an Equanimity Scale (the EQUA-S) in a Western population with or without meditation experience, based on previous definitions of equanimity, in order to investigate its relations with the relevant psychological constructs and health outcomes.

**Methods:** Adults from the general population ($N$ = 265; $M_{age}$ = 34.81) completed various measures: the EQUA-S, mindfulness, hyper-sensitivity, avoidance and fusion, impulsivity, personality, alexithymia, sensitivity to punishment and reward and frequency of problematic addictive behaviors. The dimensionality of the EQUA-S was examined using Factor Analyses. The convergent validity of this new scale was investigated using Pearson's Correlations.

**Results:** The results of a factor analysis revealed two dimensions of equanimity: an even-minded state of mind (E-MSM) and a hedonic independence (HI) component. While the E-MSM was positively related to emotional stability, adaptive emotional regulation and several mindfulness-related abilities, HI was found to correlate negatively with addictive issues.

**Discussion:** The relations with personality constructs and possible related cognitive processes are discussed.

# INTRODUCTION

Mindfulness has been defined as paying non-judgmental and non-reactive attention to the present moment. The practice of mindfulness-based meditation has been found to have a robust effect on a variety of psychological outcomes, such as changes in emotionality, relationship issues, attention and health (*Sedlmeier et al., 2012*). Several psychological and

neurological mechanisms underlying these effects have been identified (*Gu et al., 2015*). In Buddhism, mindfulness meditation is a way to achieve an attentional, emotional and cognitive balance of the mind (*Ekman et al., 2005*), a mental state which can be termed equanimity. Many authors have suggested using equanimity as a general outcome in contemplative research (*Desbordes et al., 2015*; *Hadash et al., 2016*), but existing mindfulness scales do not take this primordial quality into account (*Weber, 2017*).

In the *Abhidhamma Sangaha* (*Bodhi, 2012*), a classical Buddhist text from the Theravada tradition, and also in the Mahayana teachings on Buddhist psychology (i.e., *lorig* in the Tibetan tradition; *Gyatso, 2002*), equanimity (*Upekkhā* in Pali) is defined in various forms, including equanimity as a feeling, immeasurable equanimity and equanimity as a mental attitude. Equanimity as feeling refers to a way to neutrally experience an object, or in other words, with a "neither-painful-nor-pleasant feeling" (*Bodhi, 2012*). Immeasurable equanimity forms part of the meditation practice on the Four Immeasurables (love-kindness, compassion, joy and equanimity), which aim to develop a deep sense of compassion and care in an individual towards all living beings by iteratively familiarizing them with these four states (*Wallace, 2010*). Finally, equanimity is also defined as a balanced mental attitude or quality of mind, with unbiased reactions to things. More specifically, all objects, situations, thoughts and emotions are considered and processed evenly, manifested by an attitude of neutrality toward all stimuli. This mental attitude of equanimity is the definition used in psychology (*Desbordes et al., 2015*), and is the focus in this article. Indeed, equanimity as a quality of mind can be developed by means of mindfulness-based meditation (*Juneau, Shankland & Dambrun, 2020*), and has recently been theoretically introduced into Western psychology as a beneficial effect of this practice (*Pagis, 2015*) based on the Buddhist conceptualization (*Dambrun & Ricard, 2011*).

Equanimity has been studied and described by Buddhists because they believe that developing this quality towards objects, thoughts, feelings and living beings leads to a decrease in suffering (*duhkha* in Pali) and an increase in happiness (*sukha* in Pali). In this context, suffering refers not only to physical pain or sadness, but also to a more all-encompassing sense of continuing dissatisfaction which is caused by a self-centered perspective in which feelings and thoughts repeatedly and automatically arise, and are vividly perceived and interpreted as "real" and as part of a stable conception of self (*Dambrun & Ricard, 2011*). However, mindfulness-based meditation allows practitioners to focus their attention on each of the sensations that make up this stream (i.e., mental proliferation or rumination) and to perceive them for what they are in the present moment: mental events, rather than a fixed reality (*Holzel et al., 2011*). With practice, the conception of the self may change, the flow of these mental events may become less automatic and habitual reactions may appear less frequently. Thus, the Buddhist conception of happiness, which involves "mental balance and insight into the nature of reality" (*Ekman et al., 2005*), can arise when an individual is free from these frustrations. Craving, for example, is one of the principal causes of frustration in Buddhist theory, and is conceptualized based on similar constructs to those of its Western definition. However, while the Buddhist conception of craving includes all afflictive attachments (e.g., striving to achieve a promotion, struggling to stop thinking about someone; *Groves & Farmer, 1994*),

Western psychology defines craving more specifically as an intense desire directed toward objects or situations, resulting in addictive behaviors (*Skinner & Aubin, 2010*). Indeed, various studies have consistently shown a decrease in addictive craving after mindfulness-based meditation (*Lacaille et al., 2014*). Thus, the development of equanimity could explain positive effects on addictive behaviors after practices, as well as a large range of positive outcomes such as prosocial attitudes (*Hadash et al., 2016*; *Romm, 2007*; *Weber, 2017*).

Equanimity can be defined as a calm and stable attitude, free of tortuous emotional reactions. This echoes the approach adopted by *Desbordes et al. (2015)*, who defined equanimity as "an even-minded mental state or dispositional tendency toward all experiences or objects, regardless of their affective valence (pleasant, unpleasant or neutral) or source" (p. 6). *Vago & Silbersweig (2012)* used *Buddhaghosa's (1991*; see also *Ortner, Kilner & Zelazo, 2007*) definition of equanimity, which refers to "a balance of arousal without hyperexcitability or fatigue" (p. 2). According to this definition, equanimity involves lower emotional interference (*Ortner, Kilner & Zelazo, 2007*), greater emotional stability (*Taylor et al., 2011*), greater inner peace (*Dambrun et al., 2012*), and reduced general stress (*Grossman et al., 2004*). When adopted in stressful situations, equanimity makes it possible for a person to remain calm and to make decisions and follow behaviors that are the least contaminated by stress and arousal as possible.

Equanimity can also be considered in terms of a motivational approach (*Hadash et al., 2016*). Hadash and colleagues used *Olendzki (2006)* definition of equanimity: "an intentional stance to neither hold on to pleasant experience nor push away unpleasant experience" (*Hadash et al., 2016*). They proposed the *Decoupling Model of Equanimity*, which conceptually defines equanimity as the decoupling of desire (i.e., wanting or not wanting) from the hedonic tone of experience. Here, the hedonic tone refers to the evaluation of the pleasantness of an object or situation, and can be understood as the valence of stimuli. Similarly, *Vago & Silbersweig (2012)* described equanimity as "impartiality without bias or discrimination arising from a sense of detachment from the attraction or aversion to ongoing experience" (p. 2). Mindfulness has been found to decouple initial automatic approach/avoidance craving reactions from their hedonic tone (e.g., alcohol; *Ostafin, Bauer & Myxter, 2012*), because according to this definition of equanimity, the approach/avoidance reaction is decreased (*Papies, Barsalou & Custers, 2012*).

As we have focused on equanimity as a quality of a balanced mind, the description of equanimity as an even-minded state of mind and as the decoupling of desire from hedonic tone also seems to encompass other definitions of equanimity. As was suggested above, equanimity and mindfulness appear to be positively and significantly related to each other, without being synonymous (*Desbordes et al., 2015*). Mindfulness has been described as the process or ability of, paying attention to moment-by-moment experience (*Kabat-Zinn, 1990*), but its definition, mechanisms, and components are much broader, leading to a wide range of operationalization (*Chiesa, 2013*; *Mikulas, 2011*). Equanimity is a quality that may be developed through mindful attention, but to date, studies have been more interested in showing the different consequences of mindfulness on emotion regulation in
general. Indeed, mindfulness-based meditation has been linked to better emotional regulation (*Kumar, Feldman & Hayes, 2008*; *Robins et al., 2012*), lower emotional reactivity (*Farb et al., 2010*), higher positive states and lower negative states in response to stimuli (*Erisman & Roemer, 2010*), improved emotional stability (*Lee et al., 2015*; *Taylor et al., 2011*), and a decrease in individuals' subjective ratings of their emotional reactions towards positive and negative stimuli (*Taylor et al., 2011*).

Noting the wide variety of outcomes and processes which have been measured so far, we propose to specifically identify equanimity as a distinct emotion regulation pattern. The identification of equanimity would allow several things. First, equanimity could help to understand observed patterns of emotion regulation that were not previously fully understood by competing models, and thus to explain several results of prior mindfulness studies (see *Juneau, Shankland & Dambrun, 2020* for a more thorough discussion of this issue). Second, the quality of equanimity could be distinguished from the mechanisms involved in the practice of mindfulness, such as decentering, non-judgment, or non-reactivity, thus enhancing the knowledge of how a state of mindfulness develops. Finally, the study of equanimity could also help to identify the benefits of sustainable happiness (*Dambrun et al., 2012*) as opposed to a fluctuating happiness based on the pursuit of pleasure which also has adverse effects on the individual (*Ryan & Deci, 2001*).

However, the literature has rarely made connections between the relevant outcomes of the mindful practice of equanimity. In order to deepen and expand the study of equanimity, it is necessary to be able to measure it. The few existing equanimity scales are based on definitions of equanimity that share some similarities, but which also contain some differences. The first of these, the Self-Other Four Immeasurables (SOFI; *Kraus & Sears, 2008*), seeks to capture participants' scores for each of the Four Immeasurables. For the equanimity subscale, participants have to rate if they thought, felt, or acted with acceptance or its opposites (i.e., indifference or apathy). Equanimity is suggested as synonymous with an acceptance of the self and of others. Moreover, the results and conclusion of the study placed greater emphasis on the distinction between positive and negative qualities related to oneself and others (*Kraus & Sears, 2008*). Considering more recent studies (*Desbordes et al., 2015*; *Hadash et al., 2016*; *Weber & Lowe, 2018*), it does not seem possible to propose equanimity as only representing a synonymous of acceptance. Indeed, acceptance is defined as "willing to experience that content fully and without defense" (*Hayes et al., 2004*, p.12) and it seems to be an important prerequisite for equanimity, allowing a person to perceive all stimuli more evenly.

In other words, acceptance is a quality of living the experience, while equanimity is a form of reactivity, with physiological (i.e., a stable and calm state of mind), emotional and motivation components. The acceptance of thoughts and emotions does not necessarily imply less intense reactions. Considering this, *Hadash et al. (2016)* suggested the use of two existing scales (i.e., anxiety sensitivity and cognitive reactivity to sadness) to assess reactivity in equanimity, and added acceptance scales to build a complete equanimity assessment. This proposal measures equanimity towards an unpleasant hedonic tone, as these authors explained. Thus, it is necessary to create a scale that also considers the responsiveness to pleasant hedonic tones.

Third, the Holistic Well Being Scale (*Chan et al., 2014*) aims to measure affliction and equanimity in a eudemonic view of well-being. It is theoretically based on spiritual care and vitality and, to our knowledge, has only been tried and validated among the Chinese population. The authors used a more global definition, of equanimity as "happiness despite an absence of pleasure" and "a state where a person abolishes his or her own sense of self" (p. 292). Equanimity items related to four factors: non-attachment (e.g., "I can accept the ups and downs in life as they come"), mindful awareness (e.g., "I am able to notice changes in my mood"), general vitality (e.g., "I am full of energy") and spiritual self-care (e.g., "I have a rich religious/spiritual life"). Considering the slight differences in definitions of equanimity, only the non-attachment questions seem to be in line with our equanimity approach.

Fourth, *Weber & Lowe (2018)* developed and validated the Equanimity Barriers Scale (EBS), which they split into four subscales (i.e., innate, interactive, reflective and social). This scale focuses on the barriers that prevent an individual from achieving equanimity by assessing their beliefs and patterns of thoughts about how judgments arise (e.g., "If my feelings change then I will change"; "I feel like the media influences the way I feel about others"). Since Weber and Lower's paper was yet to be published when we began our research, we were unable to base our work on their scale, so we independently developed the EQUA-S. Finally, the Spanish subscale of Ecuanimidad (*Moscoso & Merino Soto, 2017*), which consists of 6 items (e.g., "I feel that I am a calm person, even in moments of stress and tension"; "Stress situations emotionally disturb me" etc.), is based on the definition proposed by *Desbordes et al. (2015)*.

Considering the existing work on equanimity, our scale aimed to measure equanimity based on existing definitions and models and specifically as a quality of response to emotional stimuli with participants without meditation experience. We chose to focus on equanimity (a) as the quality of being emotionally calm and balanced, regardless of pleasant or unpleasant emotions, or in other words, as an even-minded state of mind (*Desbordes et al., 2015*) and (b) as a decrease in emotional and motivational reactions towards pleasant stimuli, due to a decoupling of desire and hedonic tone (*Hadash et al., 2016*).

We also hypothesized that equanimity can be related to fewer difficulties with emotional regulation, and that it may explain the positive effect of mindfulness-based meditation on the emotional regulatory effect of mindfulness, as well as on neuroticism, and alexithymia (*Baer, Smith & Allen, 2004*; *Giluk, 2009*; *Ostafin, Robinson & Meier, 2015*; *Siegling & Petrides, 2014*). Thus, equanimity—as a more balanced emotional reaction toward stimuli—is theorized to be negatively related to emotional negativity. Moreover, a high degree of fusion with someone's emotional state prevents flexibility and detachment with regard to such stimuli (*Corman et al., 2018*), both of which are prerequisites for equanimity. An individual's detachment from their emotional state would also reduce the frequency of impulse reactions, thus corresponding to the hedonic independence component of equanimity.

The main aim of the present study was to develop and validate a self-reported equanimity scale. We thus hypothesized the existence of two factors (even-minded state of mind and hedonic independence components), which are related to distinct psychological
**Table 1 Sample characteristics.**

| Characteristics | Sample | |
|---|---|---|
| | **A** | **B** |
| *N* | 134 | 131 |
| Age range (years) | 18–73 | 18–70 |
| Age mean (years) | 35.1 | 34.5 |
| Female (%) | 60.4 | 72.3 |
| Religious adherence (%) | 45.6 | 45.5 |
| Meditation practice (%) | 17.9 | 10.8 |

constructs. This study also aimed to investigate the possible relationships between equanimity and mental health. By reducing craving and increasing emotional regulation, equanimity, as a state of hedonic independence, can be a valuable mechanism in addressing addictive behaviors (*Berridge & Kringelbach, 2008*; *Robinson & Berridge, 2000*). We predicted that hedonic independence, which is closely related to a decrease in the approach reaction to pleasant experiences, would be negatively and significantly related to addictive behaviors. Finally, according to *Desbordes et al. (2015)*, equanimity implies non-judgment, non-reactivity, and less automatic behavior in general. Thus, we predicted that equanimity as an even-minded state would be positively associated with the non-reactive and non-judging component of mindfulness, and with efficient coping strategies (e.g., positive refocusing; see also *Jermann et al., 2006*).

## MATERIALS AND METHODS

### Participants

We recruited 265 adults in France ($N_{women}$ = 175), with ages ranging from 18 to 73 years ($M$ = 34.81, SD = 15.17). Using GPower (version 3.0.10), we estimated the required sample size for sufficient correlations power (90%). On the basis of the correlation between the FFMQ and neuroticism reported by Siegling and Petrides; $r$ = 0.47 (*Siegling & Petrides, 2014*), the minimum required sample size was calculated at 30. The participants were recruited by 150 students from the University of Clermont Auvergne. The students were asked to leave the questionnaire and consent form for 24 h in the participants' homes so that the participants could complete them while alone and undisturbed. To ensure that the questionnaire would not take too long to complete, we decided to split the participants into two samples. Both samples answered the equanimity scale, the FFMQ, and demographic questions. The participants in sample A answered the psychological constructs questionnaire, while the participants in sample B answered questions about addictive behaviors (see Table 1 for descriptions of each sample). Ethical approval for the study was granted by the Clermont Auvergne University Ethical and Research Committee (ref IRB00011450-2018621), and all procedures performed were in accordance with the 1964 Helsinki declaration. All participants provided written informed consent prior to participating in the study.

## Scale development

Based on the literature review, 42 candidate items were created to correspond to existing definitions. Some of these items were inspired by the "Ecuanimidad" subscale (*Moscoso & Merino Soto, 2017*). Three judges who were familiar with the concept of equanimity individually evaluated all these items before discussing their choices together. In the end, 25 were deemed to be sufficiently relevant to be included. Of the 25 items, 12 were designed to assess the approach to equanimity proposed by *Desbordes et al. (2015)*, which we labeled "even-minded state of mind". The remaining 13 items were inspired by the conceptualization of equanimity developed by *Hadash et al. (2016)*, which we termed "hedonic independence". The participants answered using a 5-point Likert scale (1 = *never or very rarely* to 5 = *very often or always*).

## Measures

Several constructs were measured in order to ensure the convergent validity of the equanimity scale (see Table 2). We used the available French version for each scale. We assessed mindfulness (Five Facets Mindfulness Questionnaire, FFMQ; *Heeren et al., 2011*), hyper-sensitivity (Highly Sensitive Child Scale, HSC; *Pluess et al., 2018*), the avoidance and fusion of internal events (Avoidance and Fusion Questionnaire, AFQ; *Corman et al., 2018*), impulsivity (Barratt Impulsivity Scale, BIS-10; *Baylé et al., 2000*), personality (Big Five Inventory, BFI; *Plaisant et al., 2010*), alexithymia (Toronto Alexithymia Scale, TAS-20; *Loas et al., 1996*), and sensitivity to punishment and reward (Sensitivity to Punishment and Sensitivity to Reward Questionnaire, SPSRQ; *Lardi et al., 2008*). Finally, we measured the relationships between equanimity and various health outcomes via three scales: (1) the frequency of behaviors based on a list of 16 potential addictive or problematic behaviors (e.g., video games, tobacco, etc.); (2) the frequency of eating addictions (Addictive Intensity Evaluation Questionnaire, AIEQ; *Décamps, Battaglia & Idier, 2010*), and (3) emotional regulation strategies (Cognitive Emotion Regulation Questionnaire, CERQ; *Jermann et al., 2006* and the suppression subscale of the Emotion Regulation Questionnaire ERQ; *Christophe et al., 2009*).

# RESULTS

## Exploratory factor analysis and item selection

Using SPSS Statistic 24, factor analysis with oblimin rotation of the 12 items selected to assess the even-minded state revealed a two-factor solution. The Kaiser measure of sampling adequacy (KMO) was 0.84. Based on the Eigenvalue and the Screen plot, a one-factor model appeared to provide the best fit for the data (EV = 4.28 for the first factor, and 1.42 for the second). We ran another analysis by forcing a one-factor extraction. Three items failed to load sufficiently on the first factor (i.e., a factor loading of less than 0.50). A second analysis was therefore computed with the remaining nine items. This analysis revealed a first factor solution with one item loading less than 0.50. Once this item had been withdrawn, a final analysis revealed a clear one-factor solution of eight items with a Kaiser measure of sampling adequacy of 0.85. All the items loaded appropriately on the single factor (factor loadings ranged from 0.55 to 0.72, see Table 3).

**Table 2 List of scales and subscales from sample A and B.**

| Scale | Subscale | Cronbach | Mean (SD) |
|---|---|---|---|
| Sample A | | | |
| EQUA-S | Even-minded state of mind | 0.85 | 3.10 (0.81) |
| | Hedonic Independence | 0.75 | 3.87 (0.63) |
| BFI | Extraversion (E) | 0.82 | 3.15 (0.84) |
| | Agreeableness (A) | 0.72 | 3.90 (0.57) |
| | Concientiousness (C) | 0.78 | 3.59 (0.72) |
| | Neuroticism (N) | 0.84 | 3.00 (0.93) |
| | Openness (O) | 0.73 | 3.53 (0.63) |
| HSC | | 0.70 | 5.01 (0.82) |
| FFMQ | Observing | 0.80 | 3.26 (0.81) |
| | Describing | 0.89 | 3.11 (0.90) |
| | Nonreacting | 0.77 | 3.00 (0.72) |
| | Acting with awareness | 0.86 | 3.35 (0.81) |
| | Nonjudging | 0.84 | 3.17 (0.83) |
| TAS | | 0.74 | 51.46 (11.90) |
| CERQ and ERQ | Self-blame | 0.77 | 2.49 (0.89) |
| | Acceptance | 0.66 | 3.46 (0.90) |
| | Rumination | 0.71 | 2.99 (0.91) |
| | Positive refocusing | 0.83 | 2.93 (1.04) |
| | Positive reappraisal | 0.79 | 3.63 (0.94) |
| | Refocus on planning | 0.78 | 3.53 (0.89) |
| | Putting into perspective | 0.78 | 3.70 (0.95) |
| | Catastrophizing | 0.76 | 2.03 (0.96) |
| | Blaming others | 0.77 | 1.95 (0.77) |
| | Suppression | 0.82 | 2.68 (0.85) |
| AFS | | 0.82 | 2.30 (0.50) |
| SPSRQ | Punishment | 0.88 | 2.10 (0.58) |
| | Reward | 0.83 | 2.16 (0.60) |
| BIS-10 | Motor | 0.76 | 20.64 (5.35) |
| | Cognitive | 0.37 | 25.23 (3.92) |
| | Non planning | 0.52 | 25.44 (4.48) |
| Sample B | | | |
| EQUA-S | Even-minded state of mind | 0.73 | 2.85 (0.67) |
| | Hedonic Independence | 0.73 | 3.88 (0.62) |
| HSC | | 0.67 | 5.13 (0.73) |
| FFMQ | Observing | 0.76 | 3.15 (0.78) |
| | Describing | 0.77 | 2.99 (0.68) |
| | Nonreacting | 0.63 | 2.66 (0.56) |
| | Acting with awareness | 0.86 | 3.33 (0.79) |
| | Nonjudging | 0.86 | 3.01 (0.85) |
| AIEQ | | 0.90 | |
| Frequency of addictive behaviors | | N.A | 50.38 (8.25) |

![PeerJ]

**Table 3 Factor loadings (F), means (M), standard deviations (SD), and item-total correlations (IT) for the 14 items.**

| Items | F1 | F2 | F3 | M | SD | IT |
|---|---|---|---|---|---|---|
| **Even-minded state of mind (E-MSM)** | | | | | | |
| 1. Whatever happens I remain serene | 0.72 | | | 2.90 | 1.08 | 0.17 |
|   *Quoi qu'il arrive, je reste serein(e)* | | | | | | |
| 2. I am not easily disturbed by something unexpected | 0.55 | | | 2.88 | 1.13 | 0.22 |
|   *Je ne suis pas facilement pertbé(e)ur par un imprévu* | | | | | | |
| 3. I can hardly tolerate uncomfortable emotions | −0.56 | | 0.42 | 3.11 | 1.11 | 0.34 |
|   *J'ai du mal à tolérer les émotions inconfortables (R)* | | | | | | |
| 4. I can easily get carried away by an annoyance | −0.66 | | | 3.17 | 1.2 | 0.39 |
|   *Je peux facilement me laisser emporter par une contrariété (R)* | | | | | | |
| 5. I feel that I am a calm person, even in moments of stress and tension | 0.72 | | 0.35 | 2.99 | 1.25 | 0.12 |
|   *Je ressens que je suis une personne calme, même dans des moments de stress et tension* | | | | | | |
| 6. Stress situations emotionally disturb me | −0.65 | 0.37 | 0.34 | 3.35 | 1.18 | 0.36 |
|   *Les situations de stress me perturbent émotionnellement (R)* | | | | | | |
| 7. It's hard for me to be serene during the difficult moments of everyday life | −0.66 | | | 3.20 | 1.12 | 0.37 |
|   *Il est difficile pour moi d'être serein(e) pendant les moments difficiles de la vie quotidienne (R)* | | | | | | |
| 8. I feel that the problems in my life are temporary and that they have solutions | 0.59 | | | 3.78 | 1.03 | 0.21 |
|   *Je ressens que les problèmes dans ma vie sont temporaires et qu'il existe des solutions (R)* | | | | | | |
| Full sub-scale | | | | 3.18 | 1.30 | |
| **Hedonic Independence (HI)** | | | | | | |
| 1. When I look forward to doing something pleasant, I can only think about that | 0.71 | | | 3.84 | 0.85 | 0.56 |
|   *Lorsque j'anticipe de faire quelque chose de plaisant, je ne pense qu'à ça (R)* | | | | | | |
| 2. When I anticipate a situation or something that I like, I get very excited | 0.66 | | | 3.88 | 0.88 | 0.51 |
|   *Lorsque j'anticipe quelque chose ou une situation que j'aime, je suis très excité(e) (R)* | | | | | | |
| 3. When I desire an object, I feel a strong attraction to get it quickly | 0.60 | | | 3.35 | 1.16 | 0.45 |
|   *Lorsque je suis attiré(e) par un objet qui me fait envie, je ressens une forte attraction pour l'obtenir rapidement (R)* | | | | | | |
| 4. I am very excited when I am given something pleasant (like a good surprise or a gift) or when something pleasant happens to me | 0.54 | | −0.52 | 3.94 | 0.90 | 0.37 |
|   *Je suis très excité(e) lorsqu'il m'arrive ou que l'on me donne quelque chose de plaisant (comme une bonne surprise ou un cadeau) (R)* | | | | | | |
| 5. I often wish to prolong the moments when I feel a strong pleasure | 0.60 | | | 4.32 | 0.86 | 0.47 |
|   *Je souhaite souvent prolonger les moments où je ressens un fort plaisir (R)* | | | | | | |
| 6. I can't stop doing something I like | 0.55 | | 0.35 | 3.97 | 0.92 | 0.44 |
|   *J'ai du mal à m'arrêter lorsque je fais quelque chose que j'aime (R)* | | | | | | |
| Full sub-scale | | | | 3.59 | 1.00 | |

**Note:**
(R), Reverse coded items.

The 13 items selected to assess hedonic independence were entered into a factor analysis with oblimin rotation. The KMO was 0.77, with an Eigenvalue of 3.6 for the first factor, and 1.3 for the second. Based on the Eigenvalue and the Screen plot, a one-factor model appeared to provide the best fit for the data. Seven items failed to load sufficiently on this factor (i.e., a factor loading of less than 0.50). A second analysis was computed with the remaining six items which revealed a clear one-factor solution, with all the items loading appropriately on a single factor (factor loading ranged from 0.54 to 0.71, see Table 3), and a Kaiser measure of sampling adequacy of 0.80.

## Are even-minded state of mind and hedonic independence distinct constructs?

To answer this question, we performed a new factor analysis using all the items from the two scales. Kaiser's measure of sampling adequacy was 0.80. Based on the Eigenvalues and an examination of the screen plot, this analysis revealed two factors, with the first factor accounting for 26.2% of the explained variance and comprising all the items that assess even-minded state of mind. The second factor accounted for 17.5% of the explained variance and comprising items that assess hedonic independence. Consequently, even-minded state of mind and hedonic independence were shown to be two distinct constructs. Although the two scales correlated significantly ($r = 0.174$, $p = 0.004$), the size of the correlation was small ($d = 0.35$).

The internal consistency of the two subscales was examined using Cronbach's alpha. For the even-minded state of mind, Cronbach's alpha was equal to 0.81, while for hedonic independence, it was 0.74. Thus, the two subscales of the EQUA-S had satisfactory internal consistency (see Table 2).

## Convergent validity with relevant psychological constructs

In order to assess convergent validity while controlling for each subscale of equanimity (e.g., even-minded state while controlling for hedonic independence, and vise versa), we calculated the partial correlation between the two subscales of equanimity and the relevant psychological constructs (see Table 4). In the case of the FFMQ, the even-minded state was related to nonreacting to nonjudging, and to acting with awareness. We also found partial negative correlations between the even-minded state of mind and the hyper-sensitivity score, one subscale of alexithymia (i.e., identifying emotions), and the avoidance and fusion scale. Finally, we identified a very strong negative correlation between the even-minded state of mind component and neuroticism ($\beta = -0.74$, $p = 0.000$). In order to test the robustness of the above findings, we replicated our analyses controlling for neuroticism, age, and sex. The correlation between the even-minded state of mind and nonreacting remained significant ($\beta = 0.38$, $p = 0.000$), and the correlation was still marginally significant in relation to the avoidance and fusion questionnaire ($\beta = -0.23$, $p = 0.054$). However, the correlations between the even-minded state of mind and the other components of the FFMQ (i.e., nonjudging, acting with awareness) as well as the alexithymia identifying emotions subscale and hyper-sensitivity failed to reach significance. Thus, our even-minded state subscale was found to be most closely related to nonreacting.

When the even-minded state of mind was controlled for hedonic independence was robustly and positively related to the acting with awareness component of the FFMQ. Hedonic independence was also significantly and negatively related to hyper-sensitivity, to the avoidance and fusion scale, to motor impulsivity, and to sensitivity to reward and punishment. In addition, hedonic independence was positively and significantly related to conscientiousness. When age and sex were controlled for, similar results were reached, except in the cases of the sensitivity to punishment and the conscientiousness subscale,

**Table 4 Correlations and partial correlations between Even-minded State of Mind (E-MSM), Hedonic Independence(HI), and various dependent variables.**

| | Sample | E-MSM | | HI | |
|---|---|---|---|---|---|
| | | *r* | Partial *r* | *r* | Partial *r* |
| *- FFMQ* | | | | | |
| Observing | A & B | 0.016 | 0.035 | −0.103 | −0.108 |
| Describing | A & B | 0.096 | 0.104 | −0.035 | −0.053 |
| Acting with awareness | A & B | 0.223*** | 0.193* | 0.210** | 0.179** |
| Nonjudging of inner experience | A & B | 0.296*** | 0.279*** | 0.136* | 0.090 |
| Nonreacting to inner experience | A & B | 0.540*** | 0.526*** | 0.164** | 0.084 |
| *- HSC* | A & B | −0.416*** | −0.392*** | −0.23** | −0.180** |
| *- TAS* | A | −0.102 | −0.085 | −0.105 | −0.088 |
| Identifying emotions | A | −0.203* | −0.181* | −0.150 | −0.117 |
| Describing emotions | A | −0.028 | −0.038 | 0.050 | 0.056 |
| Externally oriented thinking | A | 0.026 | 0.050 | −0.126 | −0.133 |
| *- AFS* | A | −0.435*** | −0.405*** | −0.314*** | −0.264** |
| *- SPSRQ* | | | | | |
| Sensitivity to punishment | A | −0.405*** | −0.375*** | −0.272** | −0.221* |
| Sensitivity to reward | A | −0.013 | −0.076 | −0.438*** | −0.443*** |
| *- CERQ* | | | | | |
| Self-blame | A | −0.143 | −0.124 | −0.116 | −0.092 |
| Acceptance | A | 0.327*** | 0.317*** | 0.092 | 0.035 |
| Rumination | A | −0.310*** | −0.277** | −0.254** | −0.211* |
| Positive refocusing | A | 0.151 | 0.187 | −0.166 | −0.199* |
| Positive reappraisal | A | 0.287** | 0.313*** | −0.107 | −0.169 |
| Refocus on planning | A | 0.276** | 0.330*** | −0.222* | −0.288** |
| Putting into perspective | A | 0.339*** | 0.351*** | −0.034 | −0.104 |
| Catastrophizing | A | −0.365*** | −0.345*** | −0.171* | −0.114 |
| Blaming others | A | −0.136 | −0.122 | −0.095 | −0.072 |
| *- ERQ Suppression* | A | 0.052 | 0.065 | −0.064 | −0.075 |
| *- BFI* | | | | | |
| E (Extraversion) | A | 0.016 | 0.040 | −0.13 | −0.136 |
| A (Agreeableness) | A | 0.127 | 0.101 | 0.156 | 0.136 |
| C (Concientiousness) | A | 0.117 | 0.079 | 0.225** | 0.208* |
| N (Neuroticism) | A | −0.739*** | −0.730*** | −0.181* | −0.070 |
| O (Openness) | A | 0.061 | 0.070 | 0.044 | −0.056 |
| *- AIEQ* | B | −0.118 | −0.072 | −0.304*** | −0.299** |
| *- Frequency of addictive behaviors* | B | −0.125 | −0.087 | −0.251** | −0.243** |
| *- BIS-10* | | | | | |
| Motor | A | −0.109 | −0.051 | −0.339*** | −0.327*** |
| Attentional | A | −0.185* | −0.142 | −0.279** | −0.254** |
| Non-planning | A | 0.267** | 0.260** | 0.066 | 0.018 |

Notes:
* $p < 0.05$.
** $p < 0.01$.
*** $p < 0.001$.
Partial *r* provides correlations, with the other factor of equanimity controlled for (i.e., E-MSM controlling for hedonic independence and HI controlling for E-MSM).

which failed to reach significance. Analyses with cognitive and non-planned impulsivity were not interpreted due to their low internal reliability (see Table 2).

## Relations with health outcomes

Partial correlations (see Table 4) reveal that among the emotional regulation strategies assessed by the CERQ and the ERQ, even-minded state of mind was positively and significantly correlated with adaptive regulation strategies (i.e., positive reappraisal, refocus on planning, putting into perspective and acceptance), and negatively related to inadequate strategies (i.e., rumination, catastrophizing). When we controlled for age, sex and neuroticism, the same results were found, except in the case of refocus on planning and rumination.

Hedonic independence was significantly correlated with positive refocusing, refocus on planning and rumination. When age and sex were controlled for, similar results emerged, except in the case of rumination. Finally, hedonic independence was significantly and negatively related to addictive behaviors and to problematic eating behaviors (i.e., AIEQ). We found similar results when we controlled for age and sex.

## Correlation with socio-demographic variables

We combined the two samples (A and B) into a single data set in order to measure correlations with the demographic variables. The results of a $t$-test showed a significant difference between men and women in relation to the even-minded state of mind. Women had a slightly lower score ($M = 2.81$, SD = 0.75) than men ($M = 3.28$, SD = 0.68), $t$ (262) = $-4.93$, $p = 0.001$, $d = 0.63$. We found no significant difference on the hedonic independence subscale.

A correlation analysis revealed a significant relation between the hedonic independence subscale and age ($r = 0.29$, $p = 0.001$), but no relation with age and even-minded state of mind. A $t$-test showed a significant difference, at the level of hedonic independence, between participants indicating that they had a religion ($M = 3.98$, SD = 0.55), and participants indicating that they did not ($M = 3.82$, SD = 0.64), $t$ (262) = 2.05, $p = 0.042$, $d = -0.27$. We found no correlation between profession and language, either with mindfulness frequency, even-minded state of mind, or the hedonic independence state.

## DISCUSSION

In this study, we aimed to validate a scale measuring equanimity in a population of non-meditators. Based on existing theories, we proposed two related but distinct components of equanimity: (1) even-minded state of mind and (2) hedonic independence. As was predicted, a two-factor model fitted well with the data. These two factors were positively correlated with each other, but they also shared a small amount of variance. This finding was confirmed by many distinct correlations with other measures as well as through a factor analysis. In addition, these two components of equanimity displayed adequate internal consistency.

The first component refers to equanimity as an even-minded state of mind, which means an individual staying calm and feeling low stress, irrespective of the emotional

evaluation of the situation or the stimuli. However, we found, first, that the even-minded state of mind shared a large amount of variance with neuroticism, which has been described as the opposite of emotional stability (*Gosling, Rentfrow & Swann, 2003*). Second, we identified a robust relationship between this component of equanimity and adaptive emotional regulation strategies, thus confirming that equanimity is a quality involved in emotional regulation. Third, we also found a robust relationship with nonreacting, which is a central component of mindfulness defined as letting thoughts and feelings pass without getting caught up in them (*Baer et al., 2008*). Equanimity—as an observation of someone's responses to emotional stimuli—will prevent useless and unhelpful reactions. Fourth, as expected, we also found a negative significant relationship between the even-minded state of mind and the avoidance and fusion questionnaire. Indeed, a weaker fusion with one's thoughts and feelings has been found to be related to both greater mindfulness (*Corman et al., 2018*), and greater psychological flexibility (*Hayes et al., 2006*), thus suggesting that an even-minded state of mind could be a protective health factor.

We termed the second component "hedonic independence" because it refers to the absence of actions or reactions oriented by the hedonic valence of stimuli or situations. This component was also found to correlate significantly and negatively with the avoidance and fusion questionnaire. Thus, decentering appears to be a meaningful component in developing equanimity (*Desbordes et al., 2015*). The hedonic independence component was more significantly related to addictive measures than the even-minded state of mind had been. Hedonic independence was also associated with a lower sensitivity to reward, which is a risk factor for addictions (*Dissabandara et al., 2014*; *Eichen et al., 2016*). Indeed, "approaching" or "wanting" reactions to rewarding stimuli, as in the hedonic principle, could be adaptive (*Berridge & Kringelbach, 2008*), or maladaptive by overriding the reflective system that controls a person's long-term goals (*Bechara, 2005*). These results highlight the possible absence or decrease in behavioral approach reactions to the hedonic tone of addictive stimuli. As Buddhist theories have proposed, developing equanimity can be an efficient way of reducing general craving by dampening the response to its hedonic tone.

Finally, the low correlation between the two subscales requires further investigation. This correlation could be higher on meditators because the specific practice of mindfulness increases both aspects of equanimity. In our study, the mean score was slightly higher for the HI subscale in the general population tested. It is possible that in the general population, one of the two components is more prevalent. However, additional studies are needed to understand how these two components develop over time.

Moreover, *Weber (2019)* recently described equanimity as a way to react to our inner judgements which can also extend outwards towards acceptance of the judgements of others. Inner equanimity has been defined as "open acceptance or non-reactivity towards our discrimination faculties" (ibid., p.5), while external equanimity is the acceptance of other peoples' discriminatory faculties (*Weber, 2019*; *Weber & Lowe, 2018*). In this theoretical context, the EQUA-S seems to be more oriented towards inner equanimity, and could subsequently be expanded to others as suggested by *Weber (2019)*.

Overall, the EQUA-S represents a useful tool for researchers who wish to study equanimity. We hope that this new instrument will help to stimulate research in this promising area.

### Limitations and future directions

Some limitations of this study must be acknowledged. First, our sample only concerned the general population, and we did not control for meditation experience. Nevertheless, this also highlights the relevance of the concept of equanimity and the validity of such a scale in evaluating the impact of individual differences in equanimity on health outcomes in a non-meditator population. Another recent study has examined the relation between experience in mindfulness and equanimity (*Juneau, Shankland & Dambrun, 2020*).

Moreover, the strong link between the even-minded state of mind subscale and neuroticism requires further exploration and explanation. It is possible that this relationship is connected to the dimension of emotional stability that is assessed by both the even-minded state of mind and neuroticism. If this is the case, then this relationship should disappear when a scale that more directly measures emotional stability is statistically controlled. We will explore this hypothesis in future research.

Further, this questionnaire was based on western definitions of equanimity and the authors do not claim that it measures the entire concept of equanimity. It will be useful to combine the existing scales in future, in order to explore their relationships and to improve understanding of this psychological construct. It would be interesting to examine the relations between components of the Equanimity Barriers Scale (EBS; *Weber & Lowe, 2018*) and those of the EQUA-S in a future study. The relationship between cognitive measures of emotional responses and equanimity also need further research attention. Indeed, as Buddhist theories have implied, equanimity could moderate the cognitive evaluation of emotional stimuli by promoting a more neutral evaluation. It is also possible that it leads to a decoupling of the evaluation of, and the reaction to, a stimulus. Equanimity is an important component of the mindfulness state and trait, but also of the regulation of emotion in general. This study did not address the impact of equanimity on health as a distinct process of emotional regulation, nor the development of equanimity, the barriers that inhibit it, or its relationship with mindfulness-based practice.

## CONCLUSIONS

The objective of this study was to develop a scale to specifically measure the quality of equanimity. Testing on the general population and exploratory factor analyses suggested two dimensions: an even-minded state of mind (E-MSM) dimension, and a hedonic independence (HI) dimension. The scale has good convergent validity, and its components are related to health outcomes. We therefore hope that the scale will be used in further validation studies, and that it will allow the initiation of new studies on equanimity, a promising quality that can be developed through mindfulness-based meditation.

### Funding

The authors received no funding for this work.

### Competing Interests

The authors declare that they have no competing interests.

### Author Contributions

- Catherine Juneau conceived and designed the experiments, performed the experiments, analyzed the data, prepared figures and/or tables, authored or reviewed drafts of the paper, and approved the final draft.
- Nicolas Pellerin conceived and designed the experiments, authored or reviewed drafts of the paper, and approved the final draft.
- Elliott Trives conceived and designed the experiments, authored or reviewed drafts of the paper, and approved the final draft.
- Matthieu Ricard conceived and designed the experiments, authored or reviewed drafts of the paper, and approved the final draft.
- Rébecca Shankland conceived and designed the experiments, authored or reviewed drafts of the paper, and approved the final draft.
- Michael Dambrun conceived and designed the experiments, analyzed the data, authored or reviewed drafts of the paper, and approved the final draft.

### Human Ethics

The following information was supplied relating to ethical approvals (i.e., approving body and any reference numbers):

Ethical approval for the study was granted by the Clermont Auvergne University Ethical and Research Committee (IRB00011450-2018621).

### Data Availability

The datasets analyzed for this study are available at Figshare: Juneau, Catherine; Shankland, Rebecca; Dambrun, Michael (2020): Trait and State Equanimity: The Effect of Mindfulness-Based Meditation Practice. figshare. Dataset. DOI 10.6084/m9.figshare.12162090.v3.

### Supplemental Information

Supplemental information for this article can be found online at http://dx.doi.org/10.7717/peerj.9405#supplemental-information.

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
