# Peer review of "Reliability and validity of an equanimity questionnaire: the two-factor equanimity scale (EQUA-S)"

_PeerJ, doi:10.7717/peerj.9405_

## Round 0.1 · original submission · Major Revisions

Although two reviewers find no problem with the grammar/English usage of the article I would the authors heed the advice of reviewer two and have the manuscript edited for "American English" usage - I hesitate to use the term "native English". Please pay particular attention to those suggestions made by Reviewers 1 and 2. Research is all about communication and the clearer you can be, the better your research will be received. Thank you very much for your submission, I look forward to seeing your revision.

Reviewer 1 ·

Basic reporting

This is a timely and important research topic and I commend the authors for there pursuit of this publication.

The writing is clear and relevant and the discussion around craving / attachment is significant.

There are however a number of clear oversights in this field. Firstly there has been some significant published work in the field of equanimity. Firstly an operational definition specifically tailored for Western psychology was published by Weber (2019) in clinical psychology forum. This is very important given the two facing nature of equanimity. Both 'inner' and 'outer' equanimity are clearly defined and it would seem an oversight to not include these at the forefront of discussion under the western implementation. I would expect both these definitions to be included as they formulate a timely and pertinent aspect of equanimity going forward.

Secondly, there is the Equanimity Barriers Scale that is regarded as the first scale to be published around equanimity. In an obvious narrow field of enquiry as you quite rightly highlighted, it would be remiss to not include a scale that specifically mentions the term equanimity in it. This also would seem to compliment the development of a scale to coincide with hedonic and neutral tone. This should really be considered as a scale in the methodology given the expected convergent validity. This is published under current psychology in an impact factor journal and so should be included at least in the discussion as a key aspect of secular implementation of equanimity.

Thirdly, the well established concept of 'de-centering' should be given some attention - this is possible the closes psychological component related to equanimity and so in any paper that considers equanimity as a construct should distinguish it from related constructs of non-judgement, acceptance and de-centering. The only way for a narrative that combines both Buddhist and secular mindfulness together and potentially makes it more robust is a narrative that unifies constructs/ motivations and ethical underpinning. And does not lean towards fundamentalist thinking of 'this should be the mindfulness way, this should be the Buddhist way' - what you are doing is attempting to join mindfulness more closely with the Buddhist concept of equanimity so it has to be adapted to suit the western audience. Only by complimenting both and finding unification and commonality is this narrative going to be taken seriously. Otherwise it will fall to a rejection type debate.

I commend the overall paper but for me it reads a little - this is mindfulness - this is Buddhist - but you know what is a better approach is 'look why is the arrival of equanimity pertinant to western science? To do this an oversight of an already existing western two facing definition (and that is the important aspect - as it is this that connects self to other - from inner compassion to outer compassion ) No other defintion attempts to do this - they all point to the vague nature of balanced mind, even tempered etc. This is not entirely useful to fully operationalising this construct. Then again - in western science barriers are a huge aspect of equanimity given the relative individual differences that we all have. Include these aspects and re-situate the discussion to have some more purpose the paper becomes more refined.

Experimental design

This is of a good quality and is clear with good intention.

The research questions are meaningful and rigorous investigation has certainly been implemented.

The methods are good however I do feel the oversight of the equanimity barriers scale is an issue. This should be considered.

Validity of the findings

The data appears to be highly useful and meaningful to a generalised population.

The conclusions should also link up with the re-defined introduction and tied in with were we are at with western psychology - what we need to do going forward - further studies using all existing equanimity scales in order to differentiate equanimity from what currently exists. Otherwise we are left in the same place - people say why introduce this Buddhist concept - 'de-centering' does it for you - why introduce it when acceptance and non-judgement are the same. This is a serious consideration to what you are trying to achieve.

Additional comments

Please continue this good work. I totally value this attempt and some minor work should make this more refined and useful for taking equanimity onto the next stage of research.

·

Basic reporting

- The language used throughout this paper (grammar and spelling) would benefit considerably from editing by a native speaker of English. Please see the annotated PDF for several suggestions. Please note that this is not an exhaustive list and there may be more edits required. Also, the formatting of the references needs to be checked and fixed. If APA is being used, it needs to be checked that capitalization is correctly applied.
- The introduction and background are well-structured but lack some information that would better situate the reader within the literature on equanimity and mindfulness. For example, an explanation of the difference or relationship between equanimity and mindfulness would provide greater context for some of the research cited. Similarly, it is recommended to expand more on the other current equanimity scales and their drawbacks to make a stronger case for the scale. The subscales proposed in the scale also need more explanation and better support. Please see the annotated PDF for specific comments. The research that has been referenced in the manuscript thus far, however, is well done and relevant.
- The structure of the manuscript conforms to PeerJ standards and discipline norms. It is also presented clearly.
- All the tables presented are relevant and of good quality. I would recommend checking that the titles comply with the applied formatting style. Please check that the titles and tables appear on the same page. A comment on adding information about the relationship between the outcome variables and meditation practice (in the demographic information) was also added to the annotated PDF.
- The raw data are available online and the link is provided.

Experimental design

- With regards to the research being within the scope of the journal, the study falls within the social sciences, if I am not mistaken. The journal states that “PeerJ does not publish in the Physical Sciences, the Mathematical Sciences, the Social Sciences, or the Humanities (except where articles in those areas have clear applicability to the core areas of Biological, Environmental, Medical or Health sciences).” I am unsure of how this impacts this submission.
- The research question is well defined, relevant, and meaningful. The authors make a significant contribution to the field of equanimity and mindfulness research.
- The investigation is rigorous and opens avenues for many other research opportunities. It appears that the research was conducted using high ethical standards.
- Overall, the methods have been described with enough detail. Once the language has been edited, it will enhance the clarity of this section for better comprehensibility.

Validity of the findings

- All underlying data relevant to the investigation seem to have been provided and seem robust, statistically sound, and controlled.
- Speculations are identified as such.
- Conclusions are well-stated and linked to original research question and future research directions.

Additional comments

The manuscript appears to meet the journal’s article requirements and the authors seen to have adhered to the ethical approval statement. No identifiable information was found in the data files. The experiment contributed to impelling this field of research forward and seems to have been conducted ethically.

This was a very interesting paper to review. The authors make a very important contribution to the field. I look forward to reading their revised manuscript and future research.

·

Basic reporting

This article is extremely clear with good professional English used. The references were good. The data was clear and well presented. The hypothesis was clearly stated with a good discussion of the outcome.

Experimental design

No comment. A good well designed study.

Validity of the findings

No comment.

Additional comments

I really enjoyed this paper and I think that the fact that the sample participants were non meditating adds to the very real need for the concept of equanimity and the impact it could have on mental health both generally and as part of interventions.

I wish them all luck in their next study!

---

## Round 0.2 · accepted · Accept

Thank you very much for your comprehensive efforts and paying attention to reviewers comments.